# Transferring Knowledge on Motor Development to Socially Vulnerable Parents of Infants: The Practice of Health Visitors

**DOI:** 10.3390/ijerph182312425

**Published:** 2021-11-25

**Authors:** Marlene Rosager Lund Pedersen, Marianne Staal Stougaard, Bjarne Ibsen

**Affiliations:** 1Centre for Sports, Health and Civil Society, Department of Sports Science and Clinical Biomechanics, University of Southern Denmark, Campusvej 55, 5230 Odense, Denmark; bibsen@health.sdu.dk; 2Department of Health, Social Work, and Welfare Research, UCL University College, Niels Bohrs Allé 1, 5230 Odense, Denmark; msst1@ucl.dk

**Keywords:** health visitors, socially vulnerable parents, early childhood, infants, motor development, physical activity, health promotion, implementation, co-production

## Abstract

Parents are a determinant factor in a child’s development of motor skills. Studies show that programmes in which health visitors supervise parents may improve infants’ motor skills. This study examines which factors health visitors have found to enhance and hamper the implementation of a motor development programme among socially vulnerable parents of infants. The data consist of three group interviews with 4 health visitors in each (12 health visitors in total) and a subsequent member check with 27 health visitors. All were audio-recorded and transcribed verbatim, and a thematic analysis was conducted. The results show that according to the health visitors, the programme increases the ability and willingness of parents to engage in co-producing its implementation. In particular, the materials that they hand out to the parents enhance the implementation. On the other hand, they perceive the limited time provided for the implementation, together with the many pressing needs of the families, as hampering the implementation. Consequently, the study can inform future policies and programmes for frontline workers and socially vulnerable parents of infants.

## 1. Introduction

Evidence suggests that motor skill competence is associated with physical activity [1], better respiratory fitness [2], enhanced cognitive development [3], social development [4], and language acquisition [5]. Children with better motor skills are also more likely to choose physical leisure time activities later in life [6,7,8]. Conversely, children with difficulties in motor skills are more likely to have lower self-esteem [9] and higher levels of anxiety [10]. Studies have shown that parents play an essential role in improving motor development in young children through role-modelling and providing opportunities, encouragement and support [11,12]. One way parents of infants may gain knowledge regarding the infant’s motor development is through health visitors. In Denmark, the professional expertise of a health visitor is based on 18 months of theoretical and practical training, a nursing degree, and at least two years of nursing experience with children [13]. The health visitors are a specialised professional group authorised to handle tasks related to child health. This article focuses on these health visitors and will investigate the implementation of a programme on infants’ motor development. In Denmark and other Nordic countries, the efforts of health visitors to promote the health and well-being of infants are a central part of the governments’ programmes for ensuring infants have the best start to life [14]. A key element is the Danish home visiting programme, in which the health visitors educate and supervise parents on infants’ development, including motor development. The Danish home visiting programme is very well accepted by parents [15], and only 1–2 families out of 1000 reject contact with the health visitor [16]. The Danish Health Authority has established guidelines for the number of times the health visitors have to visit families during the child’s first year [16]. However, the final amount of visits is decided by the individual municipality [16]. Early programmes and interventions, in which health professionals guide parents in exercise, have been found to positively affect children’s motor development in premature infants [17]. Programmes and interventions for infants will require parental involvement, as they are the ones who must realise daily activities to promote children’s motor development. It presupposes that the parents see the importance of emphasising the child’s motor development and putting activities into everyday life conducive to motor development. Thus, the successfulness of the health visitors in passing on their knowledge depends on the parents’ readiness and motivation to learn. The readiness of citizens to engage with and benefit from a programme has become a considerable concern to policymakers due to the need to carefully target governmental resources [18,19,20,21]. Previous studies have shown a correlation between parents’ socio-economic status or mothers experiencing stress or depression and challenges to their children’s early physical and mental development [22,23,24,25,26,27]. Infants of parents in socially vulnerable positions, therefore, require special attention. One strategy used in early programmes and interventions is the use of co-production. The literature investigating the role of citizens in implementing public services or programmes uses the term co-production to stress the active role of citizens in ensuring the quality and purpose of public services [28,29,30]. In relation to co-production, several studies have pointed out that people in socially vulnerable positions may lack the necessary resources, including skills and knowledge, to engage in co-producing the services they need [29,31,32,33,34]. Still, other studies find that, provided with the appropriate tools, information, consultation, and a safe environment, socially vulnerable citizens have been able to engage in co-producing the public services on which they depend [28,29,32,34].

As shown above, we know that parents of infants may affect the child’s motor development and that programmes where health visitors guide parents have a positive effect. However, previous studies have not investigated the experience of the health visitors in direct contact with the families. From implementation research, we know that these frontline workers are central to successful implementation [35]. Furthermore, implementation studies on health professionals show that barriers exist throughout the process when implementing new programmes [18,19,20]. Consequently, we need further knowledge on the experiences of health visitors when engaging parents in socially vulnerable positions in co-producing their infants’ physical activity and motor development. Therefore, this article investigates which factors the health visitors experience as hampering or enhancing for the implementation of a real-life field experiment programme in co-production with socially vulnerable parents of infants. The programme’s success depends on a successful implementation [35]. Therefore, this article examines the implementation of the programme. The focus will be on the implementation process, not the effect of the concrete programme. The programme consists of various elements that the health visitors must pass on to the parents of infants in the Danish home visiting programme. The programme aims to increase parents’ knowledge and action regarding motor development and motor skills. The study will contribute new knowledge to policymakers on essential matters to consider when designing a programme for health visitors to implement in co-production with parents of infants.

## 2. Aim

The study investigates the health visitors’ experience of implementing a programme on motor development in co-production with socially vulnerable parents of infants. Furthermore, the study identifies possible factors that enhance and hamper the implementation.

## 3. Research Setting: The Real-Life Experiment Programme

In 2020, the Danish municipality of Hoeje-Taastrup commenced a real-life experiment programme. The programme aims to develop the competencies of parents of infants concerning motor development. It is the responsibility of the municipality’s health visitors to implement this programme in co-production with the parents. The health visitors must provide knowledge of motor development to the parents during the mandatory visits. The health visitors have eight mandatory home visits to parents of infants, starting when the child is four to five days old (see Table 1). The same health visitor visits the same family. See Table 1 for the health visitors’ mandatory visits to the parents of infants.

The Danish home visiting programme has been implemented on a national level since 1974. As part of the home visiting programme, the health visitors go to the home of the infants and perform regular examinations during infancy. This includes measuring the growth of the head, length and weight, evaluating motor and speech development, duration of breastfeeding, guidance of infants’ emotional and developmental needs, screening the parents for postnatal depression, and guiding them in their new roles.

Of the 98 Danish municipalities, Hoeje-Taastrup Municipality was ranked in 2018 as the municipality with the 14th lowest socio-economic level, based on demographic and socio-economic predictors such as labour market attachment, education and income levels, crime levels, housing situations, psychiatric patients, and the number of residents of non-Western origin [36]. Despite a high degree of economic equality in Denmark [37], inequality and deprivation exist. This municipality is faced by a comparatively high number of residents in socially vulnerable positions, including neighbourhoods with an exceptionally high concentration of socially vulnerable residents [38]. We use the term social vulnerability to depict a position caused by a complex overlap of different socio-economic factors and societal structures. The degree to which a person is socially included or excluded is based on an interplay between social, demographic, economic, and behavioural characteristics, which are mutually connected and reinforcing [39,40,41]. Danish social research points to ethnic minorities and single mothers, particularly those without vocational training who are on social benefits, as being among the most socially vulnerable citizens in Danish society [40].

The programme consists of three parts. Firstly, the health visitors received competence development on motor development. Secondly, all new parents received a bag of motor toys from the health visitors at visit 4. Thirdly, the health visitors handed out videos on motor exercises for the parents to watch and practice with their children. These three parts and their implementation are further elaborated in Table 2.

## 4. The Theoretical Framework

In order to investigate the experience of health visitors implementing a programme in practice—together with the socially vulnerable parents—we take as a theoretical starting point ‘The integrated implementation model’ by Winter and Lehmann Nielsen [35] (see Appendix A). This model is developed to explain the process of implementing a policy design in the context of a Nordic welfare state in which the state plays a major role. The model intends to describe and analyse the process by which legislation and other policy decisions are implemented. From a procedural point of view, the implementation process is seen as a phase process that begins with establishing a political agenda in the form of political discussions and consideration. This is followed by a phase in which the policy is formulated, and proposals for solutions and public action are prepared in relation to the political problem in question. This phase ends with a political decision, e.g., in the form of legislation or a strategy. In the next phase, an output, a programme, is handed over to the citizens in the form of an appropriate exercise of authority by the frontline workers. This output is expected to influence the target group and this influence thereby becomes the outcome [35].

The integrated implementation model is based on the assumption that if the actors, such as the frontline workers (in our case the health visitors) and the target group (in our case the parents), have the abilities and willingness, and are provided the opportunities (the capacity), the implementation will be successful. The allocation of resources to implementation capacity is an important part of the policy design—in this case, the programme. One cannot expect outcomes if no funds have been set aside for this. These elements from the implementation theory are applied to the case in our article in Table 3.

### Health Visitors as Frontline Workers

The rest of the article will focus on the role of the frontline worker—the heath visitor—and concerning this, the factors influencing their interaction with the target group—the parents of infants. According to Lipsky [42], the frontline workers are the real political decision makers. They hold a central position in the implementation process, as they have direct contact with the target group and, through this, are responsible for the direct implementation of the policy design. Gaining “insight into the behaviour of the frontline workers (as well as causes and consequences of this behaviour) will therefore contribute knowledge on how we (…) in this part of the implementation process can ensure as ideal an implementation as possible” (Our translation, p. 105, [35]).

Winter and Lehmann Nielsen [35] identify several basic characteristics of the frontline worker’s job, which are essential to take into consideration when trying to understand the behaviour of frontline workers in an implementation process (including the consequences that may arise in connection with this). In the current context, we will highlight the use of discretion, implementation as co-production with the target group, and limited resources available as key characteristics of the health visitors’ job, which we elaborate on below.

The use of discretion: While working within a legal framework, frontline workers employ considerable discretion in their daily decisions and their interaction with citizens [35,42]. Discretion is an inevitable part of their practice when they—based on their professional knowledge, experience, and ethical considerations—respond to the complex situations of individual citizens [42,43,44]. However, their use of discretion is based on extensive theoretical and practical training to become a health visitor [13]. Besides this, many receive further training, such as becoming lactation consultants or various types of children’s therapists. As pointed to initially, the health workers also work according to specific instructions on when to visit the families and which topics to address. However, based on a professional assessment, they exercise discretionary judgment on how to conduct the individual visit, when and how to address the different topics, and whether a family needs extra home visits.

Limited resources available: Another basic characteristic of the frontline worker’s job is having limited resources available to them. Consequently, each frontline worker has to prioritise how to spend their time, energy, and other resources [35]. According to Lipsky (2010), political or managerial consensus on how to prioritise the limited resources rarely exists, and consequently, the frontline workers are met by different (and high) expectations alongside their own professional standards, which may result in the experience of cross-pressure. It is also within this context of limited resources that the health visitors exercise their discretionary practice.

Co-production with the target group: Implementation takes place through interaction between the frontline workers and the target group, which Winter and Lehmann Nielsen [35] refer to as *joint production.* This corresponds to the notion of co-production, which highlights the active role of citizens in ensuring the successful implementation of public services [33,45,46]. In order to engage in co-production, the target group needs the abilities (resources, knowledge, etc.) and the willingness (motivation and interest) to do so [28,29,35]. Consequently, the frontline workers’ ability to handle and engage with the target groups’ abilities and willingness, or lack thereof, has a significant impact on the implementation. For this task, the frontline workers depend both on the resources provided in the policy design and their own professional discretionary practice.

## 5. Methods

### 5.1. Participants

The participants included all health visitors of the Municipality of Hoeje-Taastrup (*n* = 27 female health visitors).

### 5.2. Data Collection

The data, consisting of three in-depth group interviews, a member check, and a written document, were gathered over six months, from January to June 2021. Informed consent was obtained verbally from all subjects involved in the study. We conducted three group interviews with four health visitors in each. Qualitative in-depth interviews are beneficial for providing access to the informants’ interpretations of events, views, experiences, and understandings, information which is difficult to obtain through formal questionnaires. Furthermore, this type of data “allows for more complex analysis” [30]. The group interviews were chosen in order for the health visitors to reflect together and present several perspectives on the same topic. The research participants were selected to represent all three municipal districts, accommodating a possible demographic variation in the municipality. The participants varied in age (from 28 to 62) and years of experience as health visitors (1–22 years). Thus, a total of 12 health visitors participated in these group interviews. They all agreed to participate, and the group interviews were conducted virtually (via TEAMS due to COVID-19), and were recorded for subsequent analytical purposes and validation. The group interviews provide a way to share their experience of disseminating the programme to the socially vulnerable parents of infants, including which factors they found to hamper or enhance implementation. The group interviews were conducted by use of a semi-structured interview guide [47], including open questions regarding the daily work of the health visitors, their home visits related to motor development, and their concrete experiences of transferring the programme to the parents. The principal investigator conducted the group interviews. Each group interview lasted 60 min and was conducted in January or February 2021. In addition, member checking was conducted [48] at a meeting with all the health visitors of the municipality (*n* = 27). At this meeting, the initial results of the analysis were presented, and the participants validated the credibility thereof. The meeting was recorded for subsequent analytical purposes. Table 4 provides an overview of the data collected.

### 5.3. Data Analysis and Interpretation

The coding of the data was performed by hand and was initially data driven. A thematic analysis was conducted as the first two authors coded the transcripts individually, and through common discussion, identified a number of overall themes such as the needs of the families, the issue of time, and the function of the materials [49]. The member checking for validation confirmed the relevance of these overall themes for the implementation. Subsequently, the implementation theory of Winter and Lehmann Nielsen [35] was found to be valuable for further interpretation of the themes, leading to the four themes introduced in the results section and their influence on the implementation of the programme. Thus, the course of the analysis has been characterised by a dialectic process between data and theory [50,51].

## 6. Results

In this section, we examine the factors that the health visitors experience as enhancing or hampering the implementation of the programme in co-production with vulnerable parents of infants. In the data, the health visitors point to four main factors, which will be elaborated on below. First, we will focus on two factors related to the ‘policy design’ and second, we will present two factors related to the ‘target group’. We will relate the influencing factors to the key job characteristics of the frontline workers presented in the theoretical framework section (cf. Winter and Lehmann Nielsen [35]; their use of discretion, implementation as co-production with the target group, and limited resources available), as these characteristics are found to be significant in the health visitors’ experience of implementing the programme.

### 6.1. The Two Main Factors of the Policy Design

In connection with the policy design (cf. Winter and Lehmann Nielsen [35]), which in this article is the actual design and framework of the programme, the health visitors highlight two factors which they experience as particularly influential on the implementation of the programme in practice with the parents. First, the time needed for implementing the programme in practice with the parents is not always available, and they perceive this as an impediment to the implementation. Second, the health visitors experience the supply of resources in the design, such as the materials (the videos and the bag they hand out to the parents), as an enhancing factor for the implementation. These two factors will be explored below.

#### 6.1.1. Lack of Resources in the Form of Time for Implementation in Practice

The health visitors point to time as a scarce resource that may affect the implementation. When talking to the health visitors about their practice, it becomes clear that they have many topics that need to be addressed during the visits. They talk about a standard package, a sort of mental checklist, which they have to deliver to the families at different times. During the visits, they address topics such as breastfeeding, development, measuring the baby, postpartum screening, and sleep. They describe the visits as compact, particularly if the families have a number of challenges that need to be taken care of, which postpones the programme’s implementation. In the following quote, a health visitor describes how she sometimes lacks the time to watch the videos with the families and, consequently, the time to implement the programme:


*“I have experienced times where I may not have had that much time to go into it. To just sit down and show them (…) this is for the smaller children and this is for the older children. Well, then it has not been applied to the same extent (…).”*
(#4)

This experience also applies to the materials in the bag, which they do not always have time to show the parents how to use in practice:


*“But it has also to do with time (…) so it has rather been a matter of prioritising the time, telling the parents that you can read it there and see how to do it”.*
(#13)

Another health visitor states that she sees a number of positive aspects of the programme, but finds its application difficult due to insufficient time:

In the design of the program, no additional time has been provided for implementing the programme in practice, which emphasises the frontline workers’ job characteristic of having limited resources available. Consequently, the health visitors have to implement the programme within the same amount of time that was at their disposal beforehand. Accordingly, they experience not having the necessary time for the implementation. According to Winter and Lehmann Nielsen [35], adding the needed resources such as time in connection with the implementation of a policy design is of great importance to achieve successful implementation.

#### 6.1.2. Resources in the Form of Materials Handed out to the Families

On the other hand, the health visitors experience that the bag of materials and the videos that they hand out to the parents enhance the implementation. Most of all, the parents are very excited about the materials and, in particular, some of the poor families without the means to buy the toys themselves appreciate the toys. One of the health visitors explains:


*“I find that many families are extremely happy, also with the bag, since there are some families whose toddlers don’t actually have any toys at all, and these become their only toys. So they are very grateful for the bag, but also just watching the videos is a great thing for them”.*
(#2)

The health visitors point out that to this socially vulnerable group of parents, many of whom do not understand Danish, the videos and stimulation materials are an essential tool. The materials are physical, the videos show which exercises the parents can do with their child at different ages, and they can watch them as many times as necessary. Thus, these material things transcend what they might otherwise experience as a language barrier.

According to the health visitors, this group of parents are often unaware of the type of motoric exercises they may do with their child and at what point in their child’s development. This often results in too little motor stimulation. Thus, the videos are important, since they show the parents what a child may do and when, and how the parents can support them.

Therefore, the health visitors find that this supply of resources, i.e., the bag and the videos, provided in the policy design is an important factor, which enhances the implementation. Nevertheless, the application of these material resources may still be impeded by the limited time available to illustrate their use.

### 6.2. The Two Main Factors of the Target Group

In connection with the target group (cf. Winter and Lehmann Nielsen [35]), the health visitors highlight ‘the pressing needs of the families’ and the parents’ ‘willingness to engage in co-production’ as factors affecting how they co-produce the implementation of the programme with the parents. The first factor is found to impede the implementation, whereas the other is found to enhance the implementation. This will be further explored below.

#### 6.2.1. The Pressing Needs of the Families

In their visits to the families, the health visitors use their discretionary practice to balance the regulations regarding all the topics, which they are required to talk to the families about, and the need to meet each family at eye level in order to identify what is the most important concern to address:


*“We have some things we need to cover, that we need to inform about and make sure that the families know (…). So we have checklists, but we just try to pretend that we don’t and then work dialogue-based (…), that is, being in dialogue with the family, learning what’s on their mind. If there are things we have not touched upon but know are relevant at this age, then we’ll make sure to get them in there”.*
(#8)

The health visitors express great awareness of meeting the families ‘where they are at’, i.e., ‘to see what is preoccupying them’. According to their experience, the needs of the individual families other than motor stimulation must be resolved before the programme can be implemented. The pressing needs of the families differ. In some families, the main concern may be breastfeeding problems, other families struggle with postpartum depression, while other families are worried about their newborn baby’s lack of sleep. According to the health visitors’ experience, a large group of families in the municipality are socially vulnerable, and this group has the greatest need for parental supervision in relation to motor development as well as the most problems in general:


*“We have a really big group, where there are a lot of social challenges, economic challenges, and a lot of ethnic families as well, so I would rather say that the big group consist of socially vulnerable families”.*
(#13)

The health visitors find that they must take care of these needs together with the family before they can implement the program, as the parents are otherwise not receptive to this:


*“But I need to take another step before I get to motor skills (…), for instance, making sure that the breastfeeding works. Ensuring that there is some help for the family as to how they can think or act differently so that they will get better (…) how they can get more sleep (…) there are some basic needs that need to be resolved before motor skills (…). Not that I set aside motor skills and say that it is not relevant, because it is relevant, but there is something else on the top of the list”.*
(#10)

By use of their discretionary practice, the health visitors thus address the most pressing concerns of the families. Then, once these have been handled, they can turn to implement the programme. Regarding discretion note that the room for discretion given to the frontline workers differs [35]. In this current case, according to the policy design, the health visitors were provided with knowledge and tools but without a standardised procedure of their exact application during the home visits (see Table 2). This may result in variation as to how the programme is implemented. Since the health visitors find that there are often other, more pressing needs than motor skills, these needs may hamper the implementation.

#### 6.2.2. Willingness to Engage in Co-Production

The health visitors find that the programme increases the interaction between them and the socially vulnerable parents of infants. They find that the programme captures the parents’ interest and, thus, stimulates their willingness to co-produce the implementation. For instance, one of the health visitors’ states that the materials are like a means to achieve the goal of reaching and motivating the parents:


*“The bag is a really good way to approach the talk about motor skills (…). The families are just so happy (…) no way, are you bringing us a gift? And I say, yes let me tell you, and then they take the things out of the bag. That is, those visual things, and it is also a little bit appealing, right? Like, take a look at this colourful ball; you can use it for this (…). So, it is a real good opportunity to talk about it, without wagging one’s finger (…) it’s just the way you get in there and motivate them.”*
(#10)

Thus, the materials have had a positive impact on the willingness of the target group to engage in co-production, which has been pointed to as central to successful implementation [33,45,46]. In addition, the health visitors find it useful that the programme involves physical materials such as the bag, which helps them avoid abstract conversations about motor development and stimulation. So, the materials make the conversation about motor development more concrete in the visits.

According to the health visitors, the parents use the different materials in between their visits and are consequently co-producing the programme as hoped. The materials have provided the opportunity to continue the conversation with the parents on the stimulation of motor skills visit after visit, and the health visitors thus find that the programme has had a real impact on the target group:


*“It comes more naturally when you ask: ‘well, how did it go using it the materials?’ the next time you visit (…) Then, when the child is four months, I think the parents have become really good at saying: ‘Look at what he can do now, now he knows how to do the things in the video, and now we do like this and this’. I actually think it [the programme] has been spot on.”*
(#4)

Thus, from the perspective of the health visitors, the materials have not only had a positive impact on the willingness of the target group by stimulating their motivation. The materials have also provided them with the ability to engage in co-producing the programme. They have been given the necessary resources by being provided with the physical materials required and by gaining knowledge of how to stimulate their children’s motor development from the conversations with the health visitors. As pointed out earlier, both the willingness and the ability of a target group to engage in co-production are of great importance.

An overview of the four main factors pointed to by the health visitors is presented in Table 5. The factors are grouped in relation to policy design and target group, cf. the phases in the integrated implementation model of Winter and Lehmann Nielsen [35].

## 7. Discussion

This article provides insight into which factors the health visitors experience to enhance or hamper their implementation of a real-life field experiment programme on motor development in co-production with socially vulnerable parents of infants. In connection with the actual design of the program, i.e., the policy design, the health visitors point to two factors that affect the implementation. They find ‘insufficient time’ as hampering the implementation, while the added resources in the policy design, in the form of the materials they hand out to the families, is an enhancing factor for the implementation. In connection with the parents (the target group), the health visitors experience that the families’ other general needs and problems hamper the implementation. Conversely, they find that the materials increase the willingness and ability of the parents to engage in co-producing the implementation, and consequently, this factor enhances the implementation. Our study thereby confirms earlier studies on co-production, highlighting the importance of providing the necessary tools and consultation, in order to engage people in socially vulnerable positions in co-producing the implementation of public services [28,29,37,52,53]. Our results show that the materials handed out make the programme more concrete and visual (many of the parents do not understand Danish language). According to the health visitors, they enhance the parents’ willingness and ability to engage in co-production. Moreover, the health visitors find that their message gets through, i.e., that the parents are receptive and motivated. The materials in our study ease the communication and make the recommendations understandable and more acceptable.

As shown in the results section and discussed above, the supply of resources, i.e., the bag and the videos, provided in the policy design is an important factor, which enhances the implementation. However, the application of these material resources may still be impeded by the limited time available to illustrate their use. According to the health visitors, the socially vulnerable families have challenges that must be addressed before the health visitors can implement the programme, and because of the limited time, this may mean that the implementation may not be as successful as it could be. Our results point to insufficient time as a hampering factor to the implementation, which is in line with previous studies finding time constraints to be a main factor for not implementing a new programme [18] and that barriers exist throughout the process when implementing new programmes [18,19,20]. The theoretical framework was helpful in our interpretation of the results because the model explains the elements necessary for successful implementation. By applying the theoretical framework, we could investigate whether and to which degree these elements were present in our study.

### 7.1. Implications for Health Promotion

This study does not measure the effect of the programme but the factors that hamper and enhance the implementation of the programme. One way to teach socially vulnerable parents of infants the importance of their infant’s motor development may be through health visitors, as shown in this article. The health visitors are in regular contact with the families, who, to a large degree, have trust in the home visiting programme [15], and this may work as a platform for upgrading the parents’ competencies and knowledge. Practitioners who design initiatives to promote infants’ physical activity and motor skills should consider the results of the study when designing health promotion strategies to socially vulnerable parents of infants. Ideally, all items in Table 5 should be taken into consideration when practitioners seek to promote a programme to qualify parents of infants regarding their infants’ motor development, such as allowing time for implementation in the policy design and prioritising materials for the target group. Firstly, one must be aware that such a programme requires sufficient resources. However, it can be challenging to compete with other programmes, where time and resources are also limited. Therefore, when launching such a programme, a general prioritisation of the various focus areas in the health visitors’ home visits needs to be made in order to ensure suitable time to promote the infants’ motor development. One way to address the problem regarding the lack of time could be the use of digital technologies to supplement the home visits, for instance by online meetings with the health visitor or the streaming of motor skill classes. However, in this case, the actual access of the vulnerable families to these technologies would need to be considered. Furthermore, the target groups’ curiosity regarding the materials could be further utilised by distributing some materials in each visit, to maintain the parents’ curiosity and open-mindedness to motor skills development. Still, this could also take time away from other important topics that need to be addressed at different visits.

Secondly, one must consider whether other actors can assist in strengthening the parents’ understanding and knowledge of the importance of children’s motor development. There are opportunities for parents and their infants to participate in baby swimming and other physically stimulating activities organised by voluntary organisations in Denmark. Experience with collective co-production, where voluntary organisations collaborate with public authorities on implementing public initiatives, could be used to develop collaboration in this area.

### 7.2. Strengths and Limitations

The study included a relatively large sample size (*n* = 27): all the health visitors in the Municipality of Hoeje-Taastrup, which we perceive as a strength. By having all districts of the municipality represented in the group interviews, a possible demographic variation would have been noted. Furthermore, the programme was a natural experiment in a real-life setting, taking the complexities of social interactions and contexts into account. This may be a strength in the transferability of the experiences to other social and complex contexts. However, the particularity of the Danish Home visiting programme presupposes a certain translation into other settings.

The focus of the article has been on the perspectives of the health visitors and consequently, the parents’ experiences with the implementation of the programme have not been examined. Therefore, we only know second-hand, from the health visitors, that the parents have been engaged in co-producing the implementation of the programme. We do not know if this has also been the experience of the parents. Supplementing the data with observation studies during home visits and interviews with parents could have helped validate the health visitors’ experience.

### 7.3. Implications for Future Research

This article is part of a more extensive study. In future studies, we will investigate the effect of the programme on the infants and the parents’ perspectives on the programme. The importance of the non-human actors in the form of the materials handed out to the parents could be further investigated by use of the actor–network theory (ANT), developed by Latour and Callon [54]. This perspective stresses that not only human beings act but also non-human things, and through reciprocal relationships between these different actors, action and various practices emerge [55].

## 8. Conclusions

The present study provides insight into the health visitors’ experience of implementing a programme on motor development in co-production with socially vulnerable parents of infants. The article identifies enhancing and hampering factors related to the policy design and the actual implementation of the programme with the socially vulnerable parents of infants. Based on these findings, we recommend that policymakers consider these factors in future health promotion programmes, which involve frontline workers and people in socially vulnerable positions.

## Figures and Tables

**Table 1 ijerph-18-12425-t001:** The health visitors’ mandatory visits to the parents of infants.

	The Health Visitors’ Mandatory Visits							
	Visit 1	Visit 2	Visit 3	Visit 4	Visit 5	Visit 6	Visit 7	Visit 8
**When the child is:**	4 to 5 days old	7 to 10 days old	3 weeks old	2–3 months	4–6 months	9–11 months	18 months	2.5–3 years

**Table 2 ijerph-18-12425-t002:** Description of each element of the programme.

The Programme		
Programme	Description of Each Element	Implementation with the Parents
**Competence development of the health visitors**	The health visitors took part in a competence development course on motor development. The motor development course consisted of six lessons, each lasting three hours. The first two times, the course content was motor skills in everyday life; the third and fourth time, the content was tumble play; and the last two times, the content was presence and calm. An expert in the field taught them. The courses consisted of a combination of theory and practice. In addition to this, there was a lesson on how to transfer this knowledge into practice with the parents to infants.	The health visitors had to provide knowledge on motor development, suggestions for activities, and inform about inhibiting factors on motor development. The exact transmission of this knowledge was carried out according to the individual health visitor’s use of discretion.
**A bag with motor toys**	Each family received a bag with motor stimulating toys, including a soap bubble, grip ball, massage ball, motor ball, and sensory scarf. The bag also contained a description of how to use the materials, including ideas for play and exercises. In addition, it included swimming tickets and a brochure on activities for infants in the municipality of Hoeje-Taastrup.	The bag was handed out to the parents in mandatory visit 4 when the infants were two to three months old (see Table 1). Each health visitor decided how to explain and show the use of the materials during the visits.
**Ten short videos**	Ten short videos (approx. three minutes per. video) were produced for the parents. Each video targeted a specific age group and theme: Strengthening your child’s motor skills through play and exercises: Children of 1–3 months, 3–6 months, 6–9 months, and 9–12 months. Rough-and-tumble play with your child: Children of 1–2 years and 2–3 years. Motor skills in your child’s everyday life: Children of 1–2 years and 2–3 years. Tranquillity and presence for children: Children of 1–2 years and 2–3 years.	Each health visitor should inform about the videos and where to find them during the visits (through a link, QR code or website).

Reference: FP, Description of the framework for the programme.

**Table 3 ijerph-18-12425-t003:** Elements from the implementation theory applied to the current case.

Elements of the Implementation Theory	Transferred to the Case of the Article
Policy design	The programme
Frontline workers	Health visitors
Target group	Parents of infants

**Table 4 ijerph-18-12425-t004:** Outline of the data collected.

Type of Data Source		Participant (Referred to as)	Obtained/Location	Date	Time
**In-depth interviews**	Group interview in district one	Health visitor (#1) Health visitor (#2) Health visitor (#3) Health visitor (#4)	Conducted virtually (TEAMS)	140121	One hour
	Group interview in district two	Health visitor (#5) Health visitor (#6) Health visitor (#7) Health visitor (#8)	Conducted virtually (TEAMS)	280121	One hour
	Group interview in district three	Health visitor (#9) Health visitor (#10) Health visitor (#11) Health visitor (#12)	Conducted virtually (TEAMS)	110221	One hour
**Presentation of preliminary results**	Member check of initial analysis of the in-depth interviews with all health visitors present (*n* = 27)	Health visitors (#13)	The town hall in Hoeje-Taastrup	030621	Three hours
**Document**	Description of the framework for the programme	(FP)	From the Municipality of Hoeje-Taastrup		

FP: Description of the framework for the programme.

**Table 5 ijerph-18-12425-t005:** The main factors related to the health visitors’ experience of implementing the program, including which elements they experience as enhancing or hampering the implementation.

	The Main Factors Related to the Health Visitors Experience	How the Factors Influence the Implementation	The Health Visitors’ Explanation
**Policy design**	Lack of resources in the form of time	Hampering	Due to all the other topics that must be covered, they do not have enough time for implementation.
	Resources in the form of materials for the families	Enhancing	The materials they hand out to the parents are a valuable and concrete resource for the implementation.
**Target group**	The pressing needs of the families	Hampering	The needs must be addressed prior to the implementation.
	Willingness to engage in co-production	Enhancing	The materials increase the ability and willingness of the target group to engage in co-producing the implementation.

## Data Availability

All data have been published as Appendix A and were collected from published articles.

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
