# Peer review of "Transferring Knowledge on Motor Development to Socially Vulnerable Parents of Infants: The Practice of Health Visitors"

_ijerph, 2021, doi:10.3390/ijerph182312425_

Round 1

Reviewer 1 Report

International Journal of Environmental Research and Public Health 2021: Transferring Knowledge on Motor Development to Socially Vulnerable Parents of Infants: The Practice of Health Visitors

The authors used qualitative interviews with health visitors to examine the implementation of a new programme for strengthening motor-development among infants in socially vulnerable families. Implementation processes are generally understudied, and I welcome this study as an example of how to gain more insight into this particular aspect of intervention research. I have a few recommendations on how to strengthen the communication of findings.

1. The Introduction presents a fair justification for the objective of the study. The text is in my opinion a bit too long and covers topics and areas which in are not necessary for understanding the aims of the study, e. g. texts related to references 17-26. I do not request revision of the text, just a reflection.

The authors present the health visiting services twice, in the Introduction and Methods sections. In my opinion, it would function better to combine these two presentations as part of the Methods sections. I do not request such a revision, just a reflection and recommendation.  

Further, the authors use the term co-production which is a well-known term within tv-production but not so well known in health services research. I recommend that the authors add a definition or explain that co-production is about a meeting of minds coming together to find a shared solution. In practice, it involves people who use services being consulted, included, and working together from the start to the end of any project that affects them. I am aware of the presentation of the term in the Methods section, but the term is an important element in the problem statement and aim of the study, so I think it is relevant to explain the term in relation to the aim.

2. The Methods sections include a nice description of the intervention programme, the theoretical framework, presentation of frontline workers as the real political decision-makers, and presentation of participants, data collection and data analysis. The participants were 27 health visitors. Please explain whether this was all health visitors from this specific municipality or a sample; if sample, please explain the procedure.

3. The Results section explains the identification of main themes (main factors of the policy design; main factors of the target group). This is an interesting finding, nicely presented in the text and in Table 5. The balance between quotes and interpretation is appropriate. I have no recommendations regarding this part of the manuscript.

4. The Discussion section includes a brief summary of main findings and comparison with other studies, study limitations, and implications for practice (health promotion) and research. All this is fine, but I was surprised that the Discussion didn’t benefit from the study’s theoretical framework. It is my recommendation that the authors add a short text to explain whether the theoretical framework was helpful in interpreting the findings.

Minor issues

Many times, the authors use quotation marks or punctuation marks to mark texts which as far as I can see do not include a quoted passage (‘ ‘). A few examples:

At page 4 ´motor skills in everyday life’

At page 5 ´real political decision-makers´

At page 7 ´policy design´

At page 8 ´standard package´

At page 9 ´target group´ and ´policy design´

At page 10 'captures'

This is confusing to me and may also be confusing to other readers. Please skip or explain why some words need punctuation marks.

References 13 and 29 are incomplete.

Reviewer 2 Report

Thank you for your important work in promoting motor development in hard to reach groups. The article focuses on how health visitors can support parents to develop motor skills, by interviewing health visitors to find out what factors impact upon the implementation of a motor development programme. The potential of the programme is promising, given the uptake of health visitor programmes in Denmark.

The introduction is well written and provides sufficient background and context for the research.

The methods are mostly fine as well, there are a few areas I think need a little bit more information however.

Firstly, please provide some information on which ethics board approved this research, how the participants were approached and how consent was obtained.

Secondly, could you clarify the difference between group interviews and focus groups and if one does not exist – update throughout to the use of the more widespread term ‘focus group’

Also, I can see that you used a framework to help with the interpretation of the findings, but did you use a specific framework to analyses the data (e.g. thematic).

The findings are well presented and the quotes are appropriate and support the findings. The actual findings of the research are inline with what has been previously documented and are important, despite being predictable.

The discussion and recommendations are fine – although in section 7.1 I think it is important to clarify that this study does not measure the effectiveness of the intervention, just the barriers and enablers to implementing it.

Some minor points:

In the strengths and limitations section you say that all of the health visitors in the municipality took part, however above you say that one health visitor was unable to attend – please rectify or clarify

You use the term concrete several times throughout the paper – I am not sure it is the correct word to use, in some cases it may be better to just remove the word altogether, as far as I can tell the meaning of the sentence would remain. On line 306 however you say the materials are concrete – I presume you mean physical (as in they exist), but could be read as the toys being provided are made of concrete.

Line 329 – can you clarify what you mean by ‘at eye level’

Overall I congratulate you on the work undertaken and produced here, it will no doubt be a valuable addition to health visitor practice.

Reviewer 3 Report

I would suggest expanding the description of the Theoretical Framework to avoid that the reader necessarily has to consult the appendix. It could be useful to report examples of application of the framework in other contexts and report them in the article, at least as bibliographic references.

Reviewer 4 Report

Dear authors,

first of all I am very happy that there are programs like the one studied in this paper. Equally important is the care you take in trying to improve it. Congratulations on that.

Some notes above, although many of them limited by the method used, in any case, if it is possible to improve the document, the better.

Introduction

  • In my opinion, the aim of the study could be included in the introduction session (the first session).
  • At the beginning it seemed to me that the introduction section was too long, but I understand the need to expose all the rationale for the support that is given, because only a few countries (unfortunately) have this support.

Methods

  • My concern is related to the used tool, i.e. have the questions included in the questionnaire been validated by experts? Were all participants equally questioned? Could there have been some thought induction? Were all kind of questions open?
  • It would be more enlightening if the questions asked were described

Results

  • Line 268-269 – did they suggested some time to be “the best” or “the needed”

Discussion

  • In this session, some suggestions could be included, especially regarding the problems pointed in results, such as the contact time needed to better implement the program. For instance, is here any platform or site related to this program? It may be suggested a greater proximity and contact frequency through technologies, using more videos or even some streaming classes.

Conclusion

  • Conclusions can improve by being more incisive, i.e. clear and concise

Regards

Author Response

This manuscript is a resubmission of an earlier submission. The following is a list of the peer review reports and author responses from that submission.

Round 1

Reviewer 1 Report

The manuscript presents interesting results. A few issues, however, need to be addressed:

  • You should better elaborate aim of your study to be better understood for the potential  reader
  • You should better elaborate literature-citations in introduction and discussion. The literature should be updated.
  • Insert the sampling method, inclusion and exclusion criteria. Did you use randomization in your study group? If not please explain why.
  • It is a pity that the study group is not larger. What formula did you use to determine the sample size for your study?

Reviewer 2 Report

Review manuscript ID ijerph-1336912

Thank you for the possibility to review this manuscript. The study investigated how health visitors experienced implementing an intervention to improve motor skills among vulnerable groups. This is interesting and important, however, there are some major issues that need to be addressed before it can be considered for publication.

  • Define your intervention

What is the purpose of the intervention? What do you aim to achieve? What are you expecting to get an effect of and on? In table 3, under outcomes it states “Changing the behavior of the target groups so that fewer children experience motor difficulties”, while you in the abstract write: “could increase motor development, in the long-term help to prevent lifestyle diseases”.

Depending on what is your main outcome of the intervention? What type of intervention has shown to be effective? Dosage? When during the child’s development?

Has your intervention been tested/evaluated? Who is the target group of your intervention?

Is there a relationship between early motor development and later levels of physical activity/prevention of lifestyle diseases?

Is the intervention design ready? Have you tested it? What does parents want? How should it be delivered? What is the dosage? Is the intervention effective?

  • Gap of knowledge.

What is new with your study? What is the gap of knowledge? Time is frequently stated as the main cause for not implementing a new intervention or procedure. Johansen et al (2016) showed that time was a major limitation when implementing a standardized assessment method in regular child health services (1). References from the field of implementation should be used.

  • Theoretical framework

Theoretical framework. Have you chosen the most suitable implementation framework for your study? Check out i.e. Grol and Wensing (2). Many more exists that probably are more suitable for your study.

  • Co-production

From reading the paper the authors define co-production as how the parents are engaged/involved in motor activities in-between visits. Is this co-production or rather adherence or fidelity to the intervention? Consider the use of the word co-production, is it used correctly? How much did the parents have to say about the content and delivery of the intervention?

Have the parents been involved in designing the intervention? What does parents want?

  • Effect of the intervention

If you have not measured adherence/fidelity how can you know if your intervention is worthwhile? That it is important to invest in?

How can you evaluate the effect of your intervention if it was up to the health visitors to decide how to implement the intervention in practice (page 10, line 364-369)?

Page 13, line 522. How can you examine the effect of the intervention when you have not controlled for how the health visitors implement the intervention and how the parents adhere to the activity?

  • Language and grammar

The text would benefit from being shortened and more to the point. The text should be edited for language and grammar.

  • Introduction

What is important in the introduction that will sell your story? What is your end outcome (the outcome of your intervention) and why is this study important? Shorten the text.

Use contemporary references in the introduction. References regarding developmental coordination disorder as well as early intervention for children with cerebral palsy might be included. What intervention is known to work? However, this depends on your outcome.

  • Methods

Participants. I lack information about the target group of the intervention. Was it just vulnerable groups, and how did you define vulnerable groups? Did the health visitors provide care to families not considered as vulnerable groups? Was it a universal intervention given to all or was it targeted towards children with a motor delay? Who did the health workers give the intervention to and how did they not give it to?

Data collection. It is a little unclear how you performed the interviews. It states in the abstract that your study consists of 12 qualitative interviews, but in the methods section, you state that you performed three focus groups interviews. Please clarify. Did you use an interview guide? Describe why you chose to interview the participants per region. What are the benefits/drawbacks?

Analysis and interpretation. How did you perform the analysis, did you use any specific framework? Did you analyze the data inductive or deductive? How do you ensure the trustworthiness of you results? Please also consider a section about researcher’s reflexivity.

  • Results

Present only your results in the results section. Now a large part of your results is text that should be moved to either the introduction or the discussion, i.e. page 8, line 286-293 and page 10, line 407-414. Is the text on page 9, line 313-317, your results or a discussion?

The presented results should be relevant for the aim of the study. It should answer your research question.

Use one or a maximum of two quotations per category. They should be an illustration of your results in relation to your aim.

  • Discussion

Please discuss your most important findings and how these relate to previous research.

  • Specific comments

Health visitors should be defined when first mentioned. It is not until page 5, line 178-180 we get to know that they are nurses.

Enhancing/hampering, promoting/inhibiting, use one term, define it and stick to it.

  • Reference
  1. Johansen K, Lucas S, Bokström P, Persson K, Sonnander K, Magnusson M, et al. “Now i use words like asymmetry and unstable”: Nurses’ experiences in using a standardized assessment for motor performance within routine child health care. J Eval Clin Pract. 2016;22(2).
  2. Grol R, Wensing M. What drives change? Barriers to and incentives for achieving evidence-based practice. Med J Aust [Internet]. 2004 Mar 15 [cited 2021 Aug 12];180(6 SUPPL.). Available from: https://pubmed-ncbi-nlm-nih-gov.ezproxy.its.uu.se/15012583/

Reviewer 3 Report

Thank you for allowing me to review the manuscript titled: "Transferring knowledge on motor development to socially vulnerable parents of infants: the practice of health visitors".
Let me make some suggestions and comments about the manuscript.
First of all, the introduction should be improved. The role that parents play in the development of their children is evident, but it is not a study to be seen on the effectiveness of a parental intervention, but on the role of Health visitors on parents. The introduction should include more evidence on the role of these professionals and should be more detailed in terms of content.
In relation to the intervention carried out by the Health Visitors, the manner of execution is not very clear. The weeks are explained, and the contents are broadly explained, but this research cannot be replicated if the intervention is not fully detailed (course content, toys in the bag, video content ...). In addition, we do not know the type of families, socioeconomic and demographic factors of said families ... in summary, there is nothing described in the sample to which the intervention of the health visitors has been directed. Why are parents vulnerable?
If before COVID these visits and meetings were done in person, how has it been shown that via TEAMS it is just as effective? Is this data known?
Table 4 is not explained in the text and the reader may not understand how the 4 health visitors, during an hour on a specific day, carry out an interview in a district .... this information is confusing.
In the results literal answers are offered, but many answers from some participants (3 and 10) are exposed but no answer from others ... perhaps it may not represent the whole sample.
Few references are offered in the discussion to other studies that have done similar research or with similar results. It would be good to enrich the discussion with more research that refers to its conclusions.

Round 2

Reviewer 2 Report

Thank you for the possibility to review the revised version of the manuscript! The manuscript has improved, but there are still major issues that need to be addressed before it can be considered for publication.  

  • The text should be shortened and written more to the point. Please be consistent with what you present under the different sections, i.e. you should not discuss your results under the results section but rather in the discussion section. You do not have to report what you will present in each section or what you will discuss. Keep in mind that it is a scientific paper and not a report. Kill your darlings.
  • Write the texted in past tense.
  • Language editing is still needed.
  • Did you study the implementation of the intervention or the intervention in itself?
  • What is the intervention: what the health visitor does at the visit or what the parents do with the child? When do the parents co-produce the intervention?
  • Please described the Danish home visiting program, bear in mind that most readers are not familiar with the Danish system. I would recommend that this is described in the introduction. There is a description later in the text, which is very good, but as a reader I need to know this earlier to put your study in context. I also wonder if who the home visiting program is offered to? Only socially vulnerable parents?
  • Line 17. Did your study show that “the intervention increased the ability and the willingness of parents’ to engage in co-producing its implementation”? If yes, what this the aim of this study?
  • Your intervention should be presented in the introduction. And did you study all three parts of the intervention? Did you evaluate the course? The parents did not attend this part of the intervention. What do the videos contain, information and/or exercises?
  • Please define in the introduction your definition of co-producing the intervention.
  • Do early intervention targeting motor development/motor skills affect later levels of physical activity? Please use relevant references.
  • Line 75-79. To many repetitions.
  • Does socioeconomic status affect motor development/motor skills during the first year of life? Please use relevant references.
  • The text in table 2 can be shortened. What is the framework of implementation? Is what you describe in this column what the nurses/health workers did? What profession is the expert in the field? That families are given the bag of toys at visit 4 should be added to the text. In the table, use either visit 4 or the age, not both. I assume the videos are short, not small. Do you know if the parents watched the videos?
  • Is the intervention policy design?
  • Your expected outcome is to have fewer children with motor difficulties. How do you define motor difficulties and what theory have you based your intervention on? Is toys important for children younger than 1 years of age? Why did you choose giving toys as an intervention?
  • Is it appropriate to implement an intervention without knowing if it is effective?
  • Table 3. Is the outcome of the present study to change the behavior of the target group so that fewer children experience motor difficulties? And if this is they, when can you say that the intervention has been successfully implemented?
  • Is section 4.1. necessary? Why is line 171-174 of importance?
  • You are especially interested in the use of discretion. How is this different from other healthcare professionals? What are the risks of using discretion?
  • Under participants you state that all 28 health visitors of the municipality participated, everywhere else you write 27. Was it 27 or 28? What are participants characteristics (sex, age, experience)?
  • Please describe the member check procedure. Why are the member check referred to as one person (table 4)?
  • Why did the nurses from the same districts participate in the same interviews? Benefits? Drawbacks?
  • Is focus groups in-depth interviews? What do you mean with in-depth interviews? Did they participate in something else before? What are the benefits/drawbacks of focus groups?
  • Is table 4 needed? If yes, please shorten.
  • I suggest that the quotations are referred to by i.e. (health visitor) #1, focus group 1.
  • Should data be interpreted during analysis? How did the implementation theory by Winter and Lehmann Nielsen change your initial analysis?
  • Your result section needs to be shortened. I suggest removing everything that is discussion of your results to the discussion section. You do not have to write what you are reporting below or in the following quote.
  • Only refer to your own results in the results section and report them straight to the point.
  • Use a maximum of two quotes per result section and use the quotations as illustration. You do not have to repeat what is said in the quotation in the text. I.e. line 339-341 and the quotation line 348-353.
  • Do you have information about who the health workers provided the intervention to? When did they choose not to introduce the intervention?
  • Be consistent in the words “barriers and enablers” throughout the text (or the words you choose, enhancing/hampering).
  • Line 271. I assume that you mean for the older children, not the larger.
  • You write that many of the parents did not know Danish, were the videos translated or with subtitles?
  • In section 6.1.2. you use two quotations from the same interview, do you have quotations from the other interviews or is this just a reflection of the conversation in one group in one area? If you do not have similar quotations from the other interviews I suggest that you choose one.
  • Did the nurses only provide the intervention to socially vulnerable parents? Given that all 27-28 health works participated in the study, are all families living in the area socially vulnerable?
  • How many bags where handed out?
  • I really like the quotation on line 369.
  • Line 402-406, does this need to be explained in such detail?
  • Table 5 provides a good overview over your results, but the text should be shortened. Suggestion row 2 policy design, should …valuable and concrete resource to facilitate the implementation, or do you mean the intervention…? And do the parents co-produce the implementation or the intervention? Row 4, target group.
  • In the discussion, you need to address your own results in relation to other studies. Now this is mainly re-reporting the results. Keep to what you can say something about based on your own results.
  • Please refer to studies that show that handing out toys eases the communication (line 426-428).
  • Did your study show that intervention in the home environment including learning materials had significant effect on early childhood development (line 429-431)? The same goes for line 433-434, how can you know that it was particularly the vulnerable families that benefited from the intervention when the intervention was not provided to everyone in the same way? And how many are not socially vulnerable?
  • Line 442- 444: “It is often in families where the need for parental education…” Is this true? Please use references.
  • Line 448-451 can be removed or maybe moved to the conclusion?
  • Line 454. “…how our results may be mobilized in health promotion strategies and interventions to promote infants’ motor development, and thereby reduce the risk of motor difficulties and lifestyle diseases in the long term.” Is this true? Do early interventions have an effect of later levels of physical activity? Can you based on your study say anything about this?
  • Line 486. Please insert a reference in the sentence “this is a different way if working than many professionals are used to.” How can you know that this is true? In the whole world?
  • Would your intervention differ if it was given to families with a higher socioeconomic status?
  • Are motor difficulties distributed differently based on socioeconomic status?
  • Line 491-494, repetition of previous text.
  • Line 496, how is the Danish Home visiting program unique?
  • Line 507-511, unclear.
  • Line 514-517. How do you know that the toys were especially important? Did you try the intervention without toys? And how can you know this if you have not checked for fidelity, i.e. parents’ adherence to the intervention?
  • Conclusion: what is your take home message? If you had one power point slide and you were allowed three bullet points, what is your most important findings?

Reviewer 3 Report

The introduction has been improved and they have focused on the objective of the study, which in this case is already better defined.
But despite this, the reading of the manuscript remains confusing. The methodology is unclear and the discussion has not improved enough.

Table 4 still does not have too much relevant information, and the characteristics of the families that participated and that may condition this study, still do not appear.